# An Analysis of the Risks during Personnel Transfers between Units Operating on the Water

**Krzysztof Radwanski** [1] and **Grzegorz Rutkowski** [2,*]

1 Master Mariners Association, Al. Jana Pawla II 3, 81-345 Gdynia, Poland
2 Department of Navigation, Faculty of Navigation, Gdynia Maritime University, 81-87 Morska St., 81-225 Gdynia, Poland
* Correspondence: g.rutkowski@wn.umg.edu.pl

**Abstract:** The purpose of this article is to analyse the risk related to the transfer of personnel between units operating on the water. Although several regulations exist, there are still a high volume of noncompliant personnel and passengers transfer arrangements throughout the industry. In light of this, it is the aim of this research to critically investigate and understand what is contributing to such a high level of noncompliance. In this paper, the authors outline the industry's concern about poorly assembled ladders causing severe injuries or fatalities during personnel transfers and the preventive actions taken by various organizations. Mentioned demand is supported by the analysis of selected statistical data on maritime accidents and proposals of planned preventive actions. The study also presents an example of work risk assessment for typical personnel (pilot) boarding operations with the use of pilot ladders or other means of personnel transfer. The authors hope that the formal risk assessment (RA) presented in this paper will be the basis for the development of appropriate procedures and a checklist for each crew member involved in various transfer operations to or from the ship. As a result, the widespread use of such procedures, combined with adequate training of crews in hazard identification and risk assessment, should significantly improve overall safety in maritime transport systems.

**Keywords:** risk assessment; personnel transfer; operation on the water; maritime safety; pilot ladders; pilot boarding arrangement





## 1. Introduction with Literature Overview

The International Convention for the Safety of Life at Sea (SOLAS), 1974, more specifically, SOLAS Chapter V Regulation 24, set the minimum safety standards for boarding and landing pilots, which when followed should provide an effective safeguard against injury [1].

Although the boarding and landing of pilots is regulated by many local and international organizations (e.g., International Maritime Organization (IMO) [2]), European Maritime Safety Agency (EMSA) [3], flag state authorities [4–7], there are still a high volume of noncompliant transfer arrangements throughout the industry [8–13].

Personnel transfers using pilot ladders are currently high-risk activities, depending on the size of the ship and the height of its freeboard [5,14–17]. Fatalities have taken place when individuals have fallen to their deaths while climbing up or down pilot ladders ([8,13,18,19], etc.).

There are many different types of personnel transfer arrangement (pilot ladders, gangways, combination of side door and pilot ladder, helicopter, deck to deck, etc.). In practice, as a result of many maritime inspections based on International Maritime Pilots' Association (IMPA) [20,21], United Kingdom Maritime Pilots' Association (UKMPA) [22], Australian Maritime Safety Authority (AMSA) [4], Bahamas Maritime Authorities (BMA) [5,8], United States Coastguard (USCG) [7] and the authors' own research [10,16,23,24], it has been also noted that currently we have many different types of equipment, as well as different

procedures and guidelines [5,25–28], unfortunately not always appropriate to personnel transfer. See Figure 1.

**Figure 1.** Percentage of Noncompliance by Pilot transfer type based on IMPA Safety Campaign Analysis 2017–2021 [20]. Accessed on 7 February 2022.

Many fake ladders are still sold to unsuspecting vessels and classification societies certify noncompliant arrangements. It is worth adding here that the compliance is neither complicated nor expensive.

Pilot ladders and associated equipment must comply with international standards (SOLAS V/23 [1], IMPA [20]) and be certified and properly maintained. It is sad to say, then, that pilots are still being injured and killed whilst embarking ships.

There are no definitive statistics on boarding and landing accidents or accidents with which to compare these numbers. In this case, we can only assume that in each case of death or injury, either procedures or personnel transfer arrangements were somehow inconsistent or some way noncompliant.

According to the BMA [5], recent press releases and accident reports published by various organisations across the maritime industry reveal growing numbers of serious accidents, incidents and near misses related to incorrect pilot transfer arrangements or significant defects in associated equipment. Bahamas flag vessels make up a significant number of world fleet, and the BMA [5,8] is an important source of information.

We can assume that pilots are the personnel being transferred in most of the cases of accidents. In response to accidents related to the transfer of pilots, the IMPA announced a safety campaign in 2017 [20]. It was decided that due to the increasing number of accidents, incidents and near misses related to incorrect pilot transfer, the pilot transfer arrangements would be thoroughly investigated. It was recommended that all states adopt a responsible approach based on proven safety strategies in establishing their own regulations, standards and procedures with respect to pilotage and personnel transfer.

Similarly, the BMA published in 2021 Technical Alert 21-09 [5] about their Concentration Inspection Campaign (CIC) for Pilot Transfer Arrangements on all Bahamian flagged ships. The CIC took place from 1 July 2021 to 31 December 2021. The flag approved nautical inspectors (ANIs) conducting their CICs of flag state inspections were asked to complete an additional 3-part checklist of 45 items for pilot ladder construction, rigging, associated equipment and access to the ship's deck [5].

The checks undertaken are designed to ensure regulatory compliance and that highlighted issues and requests made by the BMA have been redressed onboard their flagged vessels. In addition, all ANIs have been invited to take an online course on the Bahamas Maritime Authority pilot ladder with tips for catching up on pilot ladder safety. The response by the BMA described in the article is a good example of action implemented to reduce injuries or fatalities connected to pilot ladder and personnel transfers.

This article highlights the obligation of shipowners, operators, masters and crews to ensure that safe arrangements are in place when embarking or disembarking a vessel using a pilot ladder or another means of personnel transfer. The industry responses discussed in the article should significantly improve the safety of personnel transfer. The primary aim

of this research is to provide verifiable data which can be used to understand and improve the boarding and landing of personnel.

## 2. Methodology

The literature review highlighted a distinct lack of any research surrounding personnel injuries and fatalities, at the hands of noncompliant transfer arrangements or any research which investigated why noncompliant transfer arrangements are so prevalent in the industry. The problem is also that the literature review could not always be directly related to the boarding and landing of personnel on board a ship. To this end, mixed methodology was reviewed to further investigate and document the areas affecting compliance [3,9,12,29]. At this stage, we had to refer to accidents reported by maritime pilots that are well documented and try to do the same when transferring non-pilots (mainly crew and passengers).

Another problem is that the data collected by EMSA [3] have methodological limitations; their publication contains statistics on marine casualties and incidents which involve ships flying a flag of one of the EU member states or operating within EU member states' territorial sea or internal waters or involving other substantial interests of EU member states. Therefore, these data do not include all random events related to maritime accidents occurring in the territory of Europe. The situation is similar with other local statistical offices.

In this case, the authors adopted the following research methodology. In chapter 1, they focused on the literature review, taking into account the applicable legal regulations (SOLAS, IMPA, BMA guidelines, etc.). The statistics on maritime accidents related to the transfer of people to or from ships were analysed and described in chapter 3. Many years of sea experience in command allowed the authors to look critically at these data and add current practices applied for various types of ships.

Chapters 4 and 5 were devoted to discussing the formal risk assessment associated with the typical transfer of the crew using gangways and/or pilot ladders. In these studies, the authors used their own professional experience gained while commanding ships in the international fleet and observations from numerous audits and maritime inspections, including flag state inspections commissioned by the Bahamas, Malta and/or Barbados. The studies also took into account the existing and sometimes supplementary rules and regulations implemented by shipowners and/or ship charterers.

As a result of the research, the authors proposed a formal risk assessment presented in the last chapter. The authors hope that the data presented in this study will contribute to the development of appropriate checklists and procedures and will intensify the discussion on work safety on board in sea transport.

## 3. Analysis of Personnel Transfer Accidents

According to the UKMPA (2020) [22], in the period 2016–2020, six maritime pilots lost their lives boarding and landing vessels, and the American Club [14] predicts that pilot fatalities are as high as two or three per year.

In August 2021, the AMSA [4] received notification of the death of a crewmember who fell into the water while climbing down the pilot ladder of a bulk carrier to board a crew transfer boat. On 5 August 2020, the USCG [7] reported that a marine pilot fell from a pilot ladder and was fatally injured. The fall was the second fatal pilot ladder accident in eight months within the same port area.

Another unfortunate example is the incident involving the Maersk Kensington (2020) [13]. The pilot tragically fell whilst trying to manoeuvre past a noncompliant trapdoor arrangement.

If we take Europe as an example, a low estimate of pilotage numbers would be at least one million acts of pilotage a year across Europe [30]. When we consider these numbers concerning pilot fatalities in Europe [9,29–32], it could be argued that boarding and landing is statistically safe. However, when we consider the fact that pilots are dying at the hands of noncompliant pilot transfer arrangements (PTAs), even if it is argued that boarding and

landing are statistically safe, there is simply no argument that a noncompliant PTA is either safe or acceptable.

Additionally, in the course of our research, we established a number of key findings such as: inadequate legal regulations and/or inability to enforce them, lack of adequate training in this area and ship design errors regarding the organization of pilot boarding.

Every year, the IMPA [20,21] publishes the results of its annual safety campaign, which can be useful as background for all personnel transfer safety.

The following analysis was made using data from the IMPA safety campaign on pilot ladders from 2017 until 2021. The data were retrieved from the IMPA site and from previous years [20]:

Table 1 summarizes number of returned observations by location.

**Table 1.** Type of defects recorded in maritime transport based on Statistical Yearbook of Maritime Economy [12].

| Type of Defect | 2017 | 2018 | 2019 | 2020 |
|---|---|---|---|---|
| Error in navigation or manoeuvring | 16 | 18 | 24 | 10 |
| Loss of control | 39 | 16 | - | 2 |
| Damage of appliances | - | 5 | 20 | 14 |
| Slipping, loss of balance, unfortunate fall, hitting | 4 | 4 | 1 | 4 |
| Bad weather conditions | - | 3 | 2 | 3 |
| Hull leakage | 5 | - | - | - |
| Mechanical defects | 4 | 1 | - | - |
| Lack of caution at work | 2 | 4 | 4 | 11 |
| Immobilization by fishing net | 1 | 1 | - | - |
| Others | - | 14 | 22 | 6 |

Note: Source: Statistical Yearbook of Maritime Economy [12]. Accessed on 30 January 2022.

Table 2 shows number of returned observations by region.

**Table 2.** Number of returned observations by region based on IMPA analysis [20] in period 2017–2021.

| Region | 2017 | 2018 | 2019 | 2020 | 2021 |
|---|---|---|---|---|---|
| Africa | 55 | 100 | 43 | 173 | 76 |
| Asia Oceania | 515 | 810 | 886 | 912 | 582 |
| Europe | 1288 | 1679 | 1743 | 1788 | 946 |
| Middle East | 0 | 79 | 4 | 31 | 48 |
| N-America | 160 | 371 | 209 | 415 | 156 |
| S-America | 901 | 1300 | 1340 | 3175 | 1514 |
| TOTAL | 2919 | 4339 | 4225 | 6394 | 3322 |

Note(s): Source: IMPA Safety Campaign Analysis 2017–2021, Data by Pilotladdersafety.com [20]. Accessed on 6 February 2022.

Table 3 shows percentages of noncompliant ladders by ship type.

In a recent survey conducted by the IMPA [20,21], noncompliant transfer arrangements ranged from 6.77% to 58.06% depending on geographical location. A notable example was Europe, where 20.49% of vessels trading in the area had noncompliant transfer arrangements.

**Table 3.** Percentage of noncompliant ladders by ship type based on IMPA analysis [20] in period 2017–2021.

| Ship Type | 2017 | 2018 | 2019 | 2020 | 2021 |
|---|---|---|---|---|---|
| General Cargo | 17% | 16% | 12% | 14% | 15% |
| Oil Tanker | 17% | 12% | 14% | 9% | 9% |
| Roro | 16% | 9% | 13% | 13% | 15% |
| Passenger | 11% | 11% | 6% | 21% | 7% |
| Container | 14% | 12% | 10% | 12% | 12% |
| Gas Tanker | 13% | % | 11% | 13% | 9% |
| Reefer | 21% | 18% | 14% | 19% | 15% |
| Fishing | 40% | 38% | 37% | 15% | 81% |
| Bulk carrier | 17% | 17% | 16% | 12% | 17% |
| Chem Tanker | 21% | 13% | 11% | 16% | 9% |
| Car carrier | 5% | 10% | 11% | 7% | 10% |
| Rig Supply Vessel | 22% | 16% | 17% | 8% | 24% |
| Other e.g., navy | 22% | 12% | 18% | 15% | 13% |

Note(s): Source: IMPA Safety Campaign Analysis 2017–2021, Data by Pilotladdersafety.com [20]. Accessed on 6 February 2022.

According to our own research based on the Statistical Yearbook of Maritime Economy [12] and data taken from the State Commission on Maritime Accident Investigation in Poland [11], we can estimate that the main reasons for accidents are lack of caution at work, slipping, loss of balance, unfortunate falls and hitting. We can assume the same reasons apply to other than pilot personnel transfer. However, we must remember that according to the data of the State Commission on Maritime Accident Investigation [11], a single accident may result from more than one cause. Based on this, it looks like the best way to improve the safety will be implementing pilot ladder safety checklists and detailed training for all personnel directly involved in transfers.

Table 4 and Figure 1 summarize percentages of noncompliant ladders by transfer type based on IMPA analysis [20] in period 2017–2021.

**Table 4.** Percentage of noncompliance by transfer type based on IMPA analysis [20] in period 2017–2021.

| Transfer Type | 2017 | 2018 | 2019 | 2020 | 2021 |
|---|---|---|---|---|---|
| Pilot Ladder | 16% | 12% | 13% | 12% | 13% |
| Combination | 21% | 16% | 15% | 14% | 15% |
| Side door and Pilot Ladder | 12% | 13% | 12% | 11% | 13% |
| Gangway | 0% | 7% | 0% | 6% | 5% |
| Helicopter | 23% | 7% | 1% | 3% | 4% |
| Deck to Deck | 13% | 17% | 12% | 12% | 4% |

Note(s): Source: IMPA Safety Campaign Analysis 2017–2021, Data by Pilotladdersafety.com [20]. Accessed on 6 February 2022.

We can also group accidents by percentage and types of defects of the bulwark and/or deck arrangements. Table 5 summarizes the percentages of noncompliant ladders by the defect type.

Table 6 shows 2919 returns from participating IMPA members which have been grouped into 5 geographical areas as from 2017, and the total noncompliance is shown as a percentage of total returns from each region.

**Table 5.** Percentage of noncompliance by type of defect based on IMPA analysis [20] in period 2017–2021.

| Type of Defect | 2017 | 2018 | 2019 | 2020 | 2021 |
|---|---|---|---|---|---|
| Pilot Ladder % | 53% | 49% | 51% | 51% | 51% |
| Bulwark/Deck % | 19% | 20% | 24% | 23% | 19% |
| Combination % | 12% | 13% | 11% | 14% | 14% |
| Safety Equipment % | 16% | 18% | 14% | 12% | 16% |
| Total | 100% | 100% | 100% | 100% | 100% |

Note(s): Source: IMPA Safety Campaign Analysis 2017–2021, Data by Pilotladdersafety.com [20]. Accessed on 6 February 2022.

**Table 6.** Number of reported marine accidents by consideration based on IMPA analysis [20].

| Country | Total Returns | Compliant | Noncompliant | Noncompliant as % |
|---|---|---|---|---|
| Africa | 55 | 43 | 12 | 21.81 |
| Asia/Oceania | 515 | 475 | 40 | 7.76 |
| Europe | 1288 | 1081 | 207 | 16.07 |
| North America | 160 | 143 | 17 | 10.62 |
| South America | 901 | 709 | 192 | 21.30 |

Note(s): Source: IMPA Safety Campaign Analysis 2017–2021 [20]. Data by Pilotladdersafety.com, accessed on 6 February 2022.

However, in practice, not all personnel transfer-related marine casualties are reported (minor marine casualties and incidents are usually concealed to avoid certain legal, administrative and financial consequences), and those that are reported are not always subjected to a detailed analysis of the effects and causes of the event. As a result, less severe personnel transfer-related marine accidents and incidents are not always reported, even though they represent a significant group in the events recorded by the IMPA. A significant number of undertaken studies are terminated without issuing a report because they resulted from renounced investigations.

According to the EMSA [3], in the 2014–2019 period, 19,418 marine casualties and incidents in Europe were reported, and only in 833 cases was an investigation launched. Many of these accidents involved events recorded during the transfer of people to the ship.

Figure 2 shows that in Europe, more than a half of the maritime accidents occur in internal waters, and 27.4% of them occur in territorial seas. Based on IMPA reports [20,21] we can expect that a significant number of the above-mentioned accidents could be connected to marine personnel transfers.

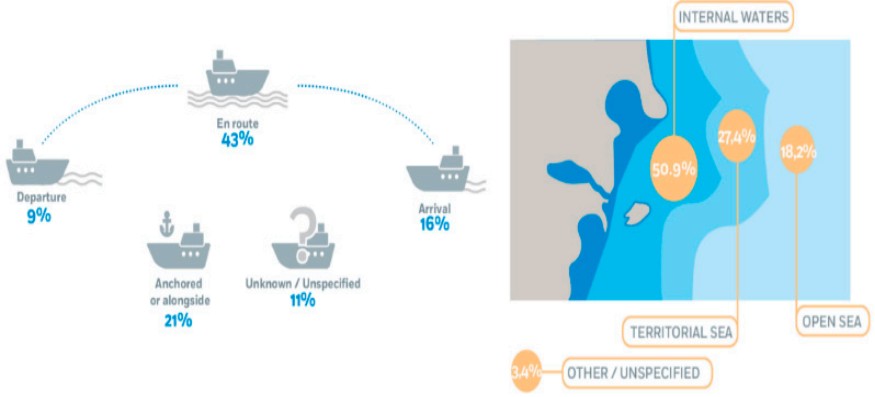

**Figure 2.** Distribution of voyage segments and location of occurrences. Source: EMSA Annual Overview of Marine Casualties and Incidents 2020 [3]. Accessed on 6 February 2022.

**4. Discussion on the Risk Assessment Related to the Transfer of Personnel to or from the Ship**

The intensity of marine traffic directly affects the safety of personnel transfer. Busy ports such as Singapore or Rotterdam need to have highly developed procedures to avoid accidents connected to personnel transfer. As shown earlier, a significant number of accidents occur in port waters [29], so inspections, prior operation checklists, safety campaigns, constant monitoring, VTS support in case of accidents and logistics that enable efficient and safe personnel transfer operations and readiness of crew are also important.

The greatest threats associated with personnel transfer always affect the fishing industry and tourism in planned heavy traffic areas [11,29,33]. Effective measures which should be taken may include improving maritime pilotage techniques; training ships' crews and shore-based personnel involved in personnel transfer operations; the stricter enforcement of regulations and above all, preparing for possible accidents through regular drills, preventive inspections and the presence of specialist rescue vessels and shore teams.

There are also several published guides to safely rigging a pilot ladder, e.g., the International Chamber of Shipping 'Pilot transfer arrangements' available from the ICS website [27], the 'Fathom Safety Guide to Pilot Ladder Securing' available from Fathom's website [25] and the New Zealand Maritime Pilot Association a 'Guide for pilot boarding operations in New Zealand when using combination arrangements with a trapdoor' available from their website [28].

It must be also noted that procedural issues can be avoided through clear communication and the implementation of requirements. Defects identified in the rigging and maintenance of ladders are generally self-explanatory and can be reduced through an effective planned maintenance regime and consistent guidance. In order to perform safety inspection to prove that pilot ladder construction is fit for carrying out personnel transfers, all detailed checklist recommendations needs to be investigated [5,21,33].

Pilot ladders may only be used for boarding and disembarking personnel [5,29]. All pilot ladders should be certified by the manufacturer and clearly identified with tags or other permanent marking so to enable the identification of each appliance for the purposes of surveys, inspection and record keeping. A record must be kept on board to confirm the date the ladder is placed into service and any repairs effected [5]. In addition, a strength test certificate for ladders must be available on board and no more than 30 months old [5]. The permanent markings showing the length to the bottom step should be provided at regular intervals (e.g., 1 m) throughout the length of the ladder.

A single length of the pilot's ladder should allow access to the water from the point of entry or exit from the ship, and all loading and trim conditions as well as unfavourable 15° list should be taken into account [5,21]. On all ships where the distance from sea level to the point of access to, or egress from, the ship exceeds 9 m, and when it is intended to embark and disembark pilots by means of the accommodation ladder, or other equally safe and convenient means in conjunction with a pilot ladder, the ship should carry such equipment on each side, unless the equipment is capable of being transferred for use on either side [5,34].

The means should be provided to ensure that any person embarking on or disembarking from the ship has safe, convenient and unobstructed passage between the top of the pilot ladder or any residential ladder and the ship's deck.

When a combination arrangement is used for pilot access, means need to be sited leading aft and provided to secure the lower platform of the accommodation ladder to the ship's side, so as to ensure that the lower end of the accommodation ladder and the lower platform are held firmly against the ship's side within the parallel body length of the ship. The pilot ladder should be able to be rigged within the horizontal distance to the lower platform between 0.1 and 0.2 m [5,21]. In addition, all means should be provided to secure the pilot ladder and manropes to the ship's side at a point of nominally 1.5 m above the bottom platform of the accommodation ladder [5]. In the case of a combination arrangement using an accommodation ladder with a trapdoor in the bottom platform

(i.e., embarkation platform), the pilot ladder and man ropes must be rigged through the trapdoor extending above the platform to the height of the handrail.

The trapdoor should open upwards and be secured either flat on the embarkation platform or against the rails at the aft end or outboard side of the platform and should not form part of the handholds.

The pilot's ladder should be clear of any possible discharges from the ship and located within the parallel body length of the ship and, as far as is practicable, within the mid-ship half length of the ship [5,27].

The pilot ladder should be secured to a strong point independent of the pilot ladder winch reel. The pilot ladder winch reel should not be relied upon to support the pilot ladder when the pilot ladder is in use. All pilot ladder winches should have mechanical means of preventing the reel from being accidentally operated as a result of mechanical failure or human error [5,21,27].

When a retrieval line is considered necessary to ensure the safe rigging of a pilot ladder, the line should be fastened at or above the last spreader step and should lead forward. The retrieval line should not hinder the pilot or obstruct the safe approach of the pilot boat.

Pilot ladders with more than 5 steps should have spreader steps of not less than 1.8 m long installed at regular intervals to prevent the ladder from twist. In addition, the lowest spreader step should be the fifth step from the bottom of the ladder and the interval between two spreader steps should not exceed nine steps. All steps should be equally spaced not less than 310 mm or more than 350 mm apart and secured in such a manner that each will remain horizontal [5,21].

The side ropes of the pilot ladder should be made of manila or other material of equivalent strength, durability and elongation characteristics. Side ropes should consist of 2 uncovered ropes not less than 18 mm in diameter on each side and should be continuous, with no joints, and have a breaking strength of at least 2400 Kg per rope [5,35]. Side ropes should each consist of one continuous length of rope, where the midpoint half-length being located on a thimble large enough to accommodate at least two passes of side rope. Each pair of side ropes should be secured together both above and below each step with a mechanical clamping device properly designed for this purpose or a seizing method with step fixtures (chocks or widgets) that holds each step level when the ladder is hanging freely.

Ladder steps, if made of hardwood, should be made in one piece and free of knots. If made of material other than hardwood, they should be of equivalent strength. All steps should be not less than 400 mm between the side ropes, 115 mm wide and 25 mm in depth, excluding any non-slip coating or grooving. No pilot ladder should have more than two replacement steps provided by the manufacturer. The four lowest steps may be of rubber of sufficient strength and stiffness. The securing strong points, shackles and securing ropes should be at least as strong as the side ropes. In addition, at the pilot's request, 2 lines of not less than 28 mm in diameter and not more than 32 mm in diameter should be properly secured to the ship [5,27]. The manropes should reach the height of the stanchions or bulwarks at the point of access to the deck before terminating at the ring plate on deck.

Personnel engaged in rigging and operating any mechanical equipment should be instructed in safe procedures to be adopted. There must also be a designated officer with the ability to communicate with the navigating bridge who will escort the pilot by a safe route to and from the navigating bridge [5].

All equipment used to transfer personnel must always be tested before use. Mechanical pilot hoists cannot be used without additional safety systems. Near the pilot ladder, there must be a lifebuoy equipped with a self-igniting light and a heaving line, separately clarified. The heaving line cannot be connected to the lifebuoy. The rope connected to the lifebuoy thrown into the water poses a potential risk of getting entangled in the ship's propellers. In all cases, adequate lighting must be provided to illuminate the transfer arrangements over side and the position on deck where a person embarks or disembarks. Shipside doors used for pilot transfer must not open outwards.

The bulwark ladder should be securely attached to the ship to prevent overturning. The bulwark ladder and two balustrade stanchions rigidly secured to the ship's structure must be fixed at or near their bases and at higher points.

Another factor that needs to be observed is footwear of each person planned to embark or disembark the vessel. Professional employees need to follow the rules and usually wear proper, industry-accepted footwear, but passengers, tourists and visitors need to be instructed or warned prior to embarkation to avoid slip, trip or fall risk. A simple poster presenting the risks connected to improper footwear should be posted. In all cases, an appropriate footwear policy should be implemented on board. Crocs, flip-flops or shower shoes, clogs with or without straps, high heels or poorly made/fitted footwear should never be used on board, especially when transferring people from or to the ship. Crew members assisting with embarkation and disembarkation should point out the risk and either ask the person involved to change their footwear or offer additional assistance during transfer. Taking into consideration the expected significant traffic increase in the coming years, it seems reasonable to have possibilities for preventing emergencies and unsafe situations.

### 5. Formal Risk Assessment for Personnel Transfer Operations between Units Operating on the Water

In maritime transport, the performance of each operation requires the preparation of an appropriate risk assessment (RA), also known as risk rating (RR). Risk rating (RR) is understood here as the product of severity (S) and likelihood (L) of the occurrence of such an event (RR = S · L). The main purpose of the risk assessment is to assess whether the work can be performed safely and what possible corrective and/or preventive actions ·need to be implemented in order for the estimated risk indicator (risk rating RR) will be considered acceptable for safe implementation of the given work [15,16,36].

Table 7 presents an example of the risk assessment with risk rating for typical boarding (disembarkation) operations of the ship's personnel (pilot) with the use of pilot ladders, gangways and/or personal transfer baskets.

The top of the Table 7 shows the analysis of the expressions used to calculate and describe the risk rating (RR) and risk assessment (RA) for person-to-ship transfer operations. The columns on the left side of Table 7 describe the hazard description, risk rating and risk category. Low-risk operations are marked in green. Medium-risk operations are marked in yellow. Operations identified as dangerous with potentially high risk are marked in red. On the right, Table 7 shows examples of control measures and/or activities to be taken to reduce risk and calculate the residual/final risk rating. RA described in this table applies to personnel transfer performed between two dynamic positioned vessels in two different cases. The first part of the table applies to general safety precautions; the second part applies to personnel transfer by basket, which is riskier and needs to be performed with extra precautions and only if absolutely necessary. The authors hope that the data presented in this study, and in particular in Table 7, will contribute to the development of appropriate checklists and implementing procedures that will intensify the discussion on work safety on board in maritime personnel transport.

**Table 7.** Formal risk assessment for person-to-ship transfer operations. Source: Own study based on the analysis of ship risk management systems used in maritime transport and the authors' maritime experience on the ships of the international fleet.

| Risk Rating | Likelihood (L) | | | | | | Analysis of expressions used to calculate risk rating RR for personnel (pilot) embarking/disembarking operation. Risk Rating RR= S·L | | | | |
|---|---|---|---|---|---|---|---|---|---|---|---|
| | | 1 | 2 | 3 | 4 | 5 | S | Severity | L | Likelihood | Risk rating (S·L) |
| Severity (S) | 1 | 1 | 2 | 3 | 4 | 5 | 1 | Negligible | 1 | Improbable | L | Low |
| | 2 | 2 | 4 | 6 | 8 | 10 | 2 | Minor | 2 | Remote | M | Medium |
| | 3 | 3 | 6 | 9 | 12 | 15 | 3 | Significant | 3 | Possible | H | High |
| | 4 | 4 | 8 | 12 | 16 | 20 | 4 | Critical | 4 | Likely | If below measures fail, please refer to Contingency Planning | |
| | 5 | 5 | 10 | 15 | 20 | 25 | 5 | Catastrophic | 5 | Certain | | |

| Hazard Description | | Risk Rating | | | Risk Category L/M/H | Control measures to be taken (in order to Reduce the Risk and calculate the residual/Final Risk Rating) | Risk Rating | | | Risk Category L/M/H |
|---|---|---|---|---|---|---|---|---|---|---|
| Consequence | Hazard | S | L | RR | | | S | L | RR | |
| 1. Personal Injury: - Fall injury - Slip injury - Impact injury- Drowning, etc. | Slip, trip & fall when moving around | 4 | 3 | 12 | Medium | Slip, trip and fall focus, staircase code, correct PPE, footwear, trained and competent personnel, work & rest hours, TBT/PTW, communication | 4 | 1 | 4 | Low |
| | Over-side movement/climbing | 5 | 4 | 20 | High | Slip, trip and fall focus, correct PPE, trained and competent personnel, work & rest hours, communication, life vest, pilot ladder, TBT/RA | 5 | 2 | 10 | Medium |
| | Pilot vessel alongside (trapped, squeezed) | 5 | 4 | 20 | High | Correct PPE, trained and competent personnel, work & rest hours, communication, life vest, pilot ladder, TBT/RA | 5 | 2 | 10 | Medium |
| | Pilot ladder failure | 5 | 4 | 20 | High | Certified pilot ladder, regular inspection of ladder, preuse inspection, protected storage | 5 | 2 | 10 | Medium |
| | Unfamiliar with operation | 5 | 4 | 20 | High | Slip, trip and fall focus, correct PPE, trained and competent personnel, work & rest hours, communication, life vest, pilot ladder, TBT/RA | 5 | 2 | 10 | Medium |
| | Long working hours/fatigue | 3 | 3 | 9 | Medium | Trained and competent personnel, work & rest hours, communication, watch handover, always four officers on watch during operations | 3 | 1 | 3 | Low |
| | Incompetent personnel to perform tasks | 3 | 3 | 9 | Medium | Competence matrix, trained and competent personnel, familiarisation checklist, two officers on duty | 3 | 2 | 6 | Medium |
| | Weather forecast—heavy weather | 3 | 3 | 9 | Medium | Weather service, predeparture checklist, sea fastening, communication, route planning/passage planning | 4 | 2 | 8 | Medium |
| 2. Ship Operations: - Unsafe operations - Navigational errors - Delayed operations | Personnel not familiarized | 3 | 3 | 9 | Medium | Trained and competent personnel, familiarisation checklist, two officers on duty | 3 | 1 | 3 | Low |
| | Weather forecast—heavy weather | 4 | 3 | 12 | Medium | Weather service, pre departure checklist, sea fastening, communication, route planning/passage planning | 4 | 2 | 8 | Medium |
| | Long working hours/fatigue | 3 | 3 | 9 | Medium | Trained and competent personnel, work & rest hours, communication, watch handover | 3 | 1 | 3 | Low |
| | Incompetent personnel to perform tasks | 3 | 3 | 9 | Medium | Competence matrix, trained and competent personnel, familiarisation checklist, two officers on duty | 3 | 2 | 6 | Medium |
| 3. Damage to Property/Asset - Collision, - Impact with, - Grounding etc. | Incompetent personnel to manoeuvre vessel | 4 | 4 | 16 | High | Competence matrix, trained and competent personnel, familiarisation checklist completed, two officers on duty, training form completed | 4 | 2 | 8 | Medium |
| | Navigational error, poor look out | 4 | 4 | 16 | High | Competence matrix, trained and competent personnel, familiarisation checklist completed, two officers on duty, training form completed | 4 | 2 | 8 | Medium |
| | Reduced Visibility—Fog, storm, snow, etc. | 4 | 3 | 12 | Medium | Competence matrix, trained and competent personnel, familiarisation checklist completed, two officers on duty, radars, digital charts | 4 | 2 | 8 | Medium |
| | Lack of communication | 4 | 3 | 12 | Medium | Establish communications, officer in charge, bridge, engine, offshore installation, crane, deck, portable radios, batteries, spare batteries, PM system, handover checklist completed before handing over the watch | 4 | 1 | 4 | Low |
| | Entry of safety zone for personnel transfer | 4 | 3 | 12 | Medium | DP checklists, toolbox talk & pre 500 m entry checklist completed before entry, permission from installation given prior to entry, certified, trained and experienced crew, emergency steering drills, evaluate and monitor tide, wind, weather conditions, extreme caution when working weather side, DP capability analysis, stop job and limit time spent alongside. If any issues or when in doubt, call Captain. | 4 | 1 | 4 | Low |
| | Loss of position reference systems PRS | 4 | 3 | 12 | Medium | Training and knowledge of PRS, onboard manual training, OOW prepared to take manual override/control, stop job policy, agreed escape routes from installation, certified crew, evaluate weather conditions, emergency steering drills | 3 | 1 | 3 | Low |
| | Loss of power/thruster | 4 | 3 | 12 | Medium | Preventative maintenance, defect reports, inspection according to class, correct bus bar setting—split bus, contingency considerations for loss of propulsion unit, stop job policy. | 3 | 1 | 3 | Low |
| | Loss of position due to weather condition | 4 | 3 | 12 | Medium | Evaluate and monitor weather, wind speed, and sea state, anticipate tidal effect, DP capability analysis, rested watch crew, be aware of weather fatigue, regular breaks, approved training, monitor power and thrusters use at all times (DP class monitor) OOW prepared to take manual override/control, stop job policy. | 4 | 1 | 4 | Low |
| | Insufficient power to retain position | 4 | 3 | 12 | Medium | Stop operation on weather side and safely move away from platform/rig/berth, avoid peak loads on position keeping power generation capacity, build in safety margins of power use levels dependent on weather/vessel/load. If power requirements to maintain station exceed 45% of main propulsion or any thrusters, including shaft alternator power, the OOW must cease operations. This limit also applies to diesel electric propelled vessels. | 3 | 1 | 3 | Low |

**Table 7.** *Cont.*

| Operation: Personal Transfer Basket | | | | | | | | | | |
|---|---|---|---|---|---|---|---|---|---|---|
| 1. Ship's Operations:<br>- Unsafe operations (fatigue, weather, etc.)<br>- Navigational errors<br>- Loss of position, power, thrusters, etc. | Long working hours /fatigue | 3 | 3 | 9 | Medium | Hours of rest form/work form, working hours monitoring, always four officers on watch during operations | 3 | 1 | 3 | Low |
| | Lack of communication | 4 | 3 | 12 | Medium | Establish communications, officer in charge, bridge, engine, offshore installation, crane, deck, portable radios, batteries, spare batteries, PM system, handover checklist completed before handing over the watch | 4 | 1 | 4 | Low |
| | Entry of 500 m zone | 4 | 3 | 12 | Medium | DP checklists, toolbox talk & pre 500 m entry checklist completed before entry, permission from installation given prior to entry, certified, trained and experienced crew, emergency steering drills, evaluate and monitor tide, wind, weather conditions, extreme caution when working weather side, DP capability analysis, stop job and limit time spent alongside. If any issues or when in doubt, call Captain. | 4 | 1 | 4 | Low |
| | Loss of position reference systems PRS | 4 | 3 | 12 | Medium | Training and knowledge of PRS, onboard manual training, OOW prepared to take manual override/control, stop job policy. Agreed escape routes from installation | 3 | 1 | 3 | Low |
| | Loss of power/thruster | 4 | 3 | 12 | Medium | Preventative maintenance, defect reports, inspection according to class, correct bus bar setting—split bus, contingency considerations for loss of propulsion unit, stop job policy. | 3 | 1 | 3 | Low |
| | Loss of position due to weather conditions | 4 | 3 | 12 | Medium | Evaluate and monitor weather, wind speed, sea state, anticipate tidal effect, DP capability analysis, rested watch crew, be aware of weather fatigue, regular breaks, approved training, monitor power and thruster use at all times (DP class monitor), OOW prepared to take manual override/control, stop job policy. | 4 | 1 | 4 | Low |
| | Insufficient power to retain position | 4 | 3 | 12 | Medium | Stop operation on weather side and safely move away from platform/rig, avoid peak loads on position keeping power generation capacity; build in safety margins of power use levels dependent on weather/vessel/load.<br>If power requirements to maintain station exceed 45% of main propulsion or any thrusters, including shaft alternator power, the OOW must cease operations. This limit also applies to diesel electric propelled vessels. | 3 | 1 | 3 | Low |
| 2. Personal Injury: (fall injury, slip injury, impact, drowning, etc.) | Hazardous transfer with basket | 4 | 3 | 12 | Medium | Trained and qualified personnel, weather/wind considered, RA/TBT, good communication all parties, stop job policy, vessel in correct position and holding position, crane function tested prior to operation | 4 | 1 | 4 | Low |
| 3. Damage to Property/Asset<br>- Collision<br>- Impacts<br>- Grounding, etc. | Landing area not suitable | 4 | 3 | 12 | Medium | Designated landing area clear, landing area agreed on by all parties, landing area cordoned off with barrier tape, only personnel involved in operation in area, beware of obstacles to personnel basket, stop job policy, | 3 | 1 | 3 | Low |

## 6. Conclusions

The statistics on maritime accidents described earlier may be disturbing. Taking into consideration the expected significant traffic increase in the coming years, it seems reasonable to have possibilities for preventing emergencies and unsafe situations.

The research conducted also showed that many of the analysed accidents could have been avoided or reduced in effects if procedures regarding personnel transfer had been better. It is also worth mentioning that regardless of the region of the world from where information on accidents was collected, a significant number of them took place in internal and coastal waters limited by the depth and width of the area.

From the data of observations in the last five years, a clear downward trend is already visible. The number of deck-to-deck noncompliance incidents has fallen sharply, which may indicate that the correct use of safety equipment and handhold stanchions has increased significantly. This means that the industry responses discussed in the article significantly improve the safety of personnel transfer but continuous improvement and constant monitoring to further reduce injuries and fatalities are common industry requirement.

The best way to improve safety during personnel transfer is to introduce simple and user-friendly tools to be used by the personnel directly involved in this operation. In this case, the risk assessment presented in Table 7 seems to be the most complex and user friendly and was chosen as one of the best from many similar procedures used by various managements, companies and vessels. The risk assessment presented in Table 7 is also a combination of all the factors mentioned previously and hopefully will be widely used by industry and various authorities as the basis for their own regulations. A similar formal risk assessment for safe personnel transfer by ladder/gangway and/or personal transfer basket should be part of each ship's safety management system and be closely followed along with informal risk assessment tools such as Toolbox Talk and Take5.

Particular attention was paid to the statistical approach to the risk analysis, which shows us many serious maritime accidents related to the transfer of personnel to or from the ship. The implementation of new safety procedures forces us to conduct further research in this area.

Effective measures which should be taken may also include improving maritime pilotage techniques; training ships' crews and shore-based personnel involved in personnel transfer operations; stricter enforcement of regulations and above all preparing for possible accidents through regular drills, preventive inspections and the presence of specialist rescue vessels and shore teams. That is good background for implementing the same procedures used locally by most involved organizations. From this point of view, the need for experience transfer also seems justified.

The formal RA outlined in Chapter 5 may form the basis for the development of a checklist for crew members involved in personnel transfer operations. Further research and discussion may result in the adoption of checklists to be filled in prior to any personnel transfer. The extensive use of appropriate procedures including RA presented in this article should improve security during the transfer of personnel.

Conducting further research in this area seems to be necessary and justified. Currently, the authors plan to continue the research with the participation of several organizations involved in coastal shipping and installations offshore in order to increase the level of safety of personnel on board their units.

**Author Contributions:** Conceptualization, K.R. and G.R.; methodology, K.R. and G.R.; validation: K.R. and G.R.; formal analysis, K.R. and G.R.; investigation, K.R. and G.R.; resources, K.R. and G.R.; data curation, K.R. and G.R.; writing—original draft preparation, K.R. and G.R.; writing—review and editing, K.R. and G.R.; visualization, K.R. and G.R.; supervision, K.R. and G.R.; project administration, K.R. and G.R.; funding acquisition, G.R. All authors have read and agreed to the published version of the manuscript.

**Funding:** This research was funded from the statutory activities of Gdynia Maritime University, grant number WN/2022/PZ/07. The Gdynia Maritime University, Polish Master Mariners Association and UMS Marbalco have been actively involved in research activities, participating in conceptual research and development studies on market analysis for avoiding and preparing for possible accidents connected to personnel transfer through inspections and regular drills as well as advising on proper approaches for directly involved personnel. This research received no external funding.

**Conflicts of Interest:** The authors declare no conflict of interest.

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
