# Peer review of "An Analysis of the Risks during Personnel Transfers between Units Operating on the Water"

_water, doi:10.3390/w14203303_

Round 1

Reviewer 1 Report

Paper deals with the risk related to the transfer of personnel between units operating on the water. This problem is very important to solve, so I appreciate this topic. The paper is of good quality, but there are some deficiencies and mistakes that must be fixed before publication.

From my point of view, the references in paper are used chaotically (the introduction section if full of sources without description what those references point to. I recommend to add a separate section Literature overview.

There are also several formal mistakes (heading on the end of page - line 198, empty lines, figures that are not in good quality for reading, tables not centred and very short conclusion - also missing discussion).

Author Response

Dear Reviewer,

We would like to thank you for review with valuable comments and suggestion of our manuscript entitled “Analysis of the operational risk related to the transfer of personnel between units operating on the water”. We have included all your comments in the revised version of the paper (please see the attachment).  Please let us know of your decision at your earliest convenience.

Yours faithfully,

Authors

Reviewer 2 Report

General comments:

-        The article requires significant changes.

-        Certain parts of the article that do not belong to the topic/area of article should be significantly shortened

-        Some parts of the chapters should be separated.

-        The conclusions of the article are too general. They should be written more precisely, i.e., after the revision of the article, the authors should write clear conclusions that are the result of RA

-        Finally, does the title correspond to the content of the article? Does the article present the analysis of operational risks or RA?

 Please, find attached specific comments of review in the separate documment.

Author Response

We would like to thank you for review with valuable comments and suggestion of our manuscript entitled “Analysis of the operational risk related to the transfer of personnel between units operating on the water”.  We have made significant changes to improve the quality of our paper in line with your suggestion and our best knowledge. See the attachment with our responses to your general and specific comments and suggestions. Please let us know of your decision at your earliest convenience.

Yours faithfully,

Authors

Author Response

(The authors gave the same response as above.)

Round 2

Reviewer 2 Report

Dear authors,

After reviewing the manuscript again, I feel that you have made most of the changes requested in the review. However, before final approval, the following changes should be made:

- Lines 25 – 27: The last sentence of the abstract contains "future plans". My kind suggestion is to summarize the results of this research (as it has been advised in my previous revision) instead of your future plans

- Lines 44, 46 and 48: There are still unnecessary citations, although some of the sources have been removed. There is no reason to use such a large number of sources in just two relatively general sentences.

-  Lines 51, 74, 147, etc. The authors have mentioned that corrections regarding the use of acronyms for their first appearance in the text have been made, some of these mistakes still exist in the whole article. My kind request is to read the text carefully once again and apply those corrections for all of the acronyms use din the manuscript.

-Methodology: After reading the existing Chapter 2 it is not so clear which methodology has been used in this article. So please describe the methodology clearly and simply.

-Line 175: You've mentioned a ''number of key findings''. Where are these findings?

- Line 222: Remove (UNCLOS 1982), because the UNCLOS is not the source of the data presented in this sentence.

- Lines 404-412: You have not made any changes in this part of the text. In your answer you have stated that ''Corrections inserted as suggested above''. This is neither professional nor fair. Please make necessary changes in this part of the text. Please briefly describe the RA you have made in Chapter 7 of this article.

Best regards,

Reviewer

Author Response

Dear Reviewer, thank you for taking your time and review our manuscript again on the 11 Oct 2022. We agree with your comments and suggestion to our manuscript. We have made changes to improve the quality of our paper in line with your suggestion and our best knowledge. See the attachment with our responses to your comments and suggestions. We hope that the changes made now are sufficient and acceptable. Please let us know of your decision at your earliest convenience.

Yours faithfully, Authors
